# Chemical Defense of Yacón (*Smallanthus sonchifolius*) Leaves against Phytophagous Insects: Insect Antifeedants from Yacón Leaf Trichomes

**DOI:** 10.3390/plants9070848

**Published:** 2020-07-06

**Authors:** Kaisei Tsunaki, Masanori Morimoto

**Affiliations:** Department of Applied Biological Chemistry, Faculty of Agriculture, Kindai University, Nara 6318505, Japan; tsunaki_k0001@pref.wakayama.lg.jp

**Keywords:** Asteraceae, *Smallanthus sonchifolius* (Poepp. & Endl.) H. Rob., sesquiterpene lactones, uvedalin, enhydrin, insect antifeedants, *Spodoptera litura* F.

## Abstract

Yacón is a perennial crop with high insect resistance. Its leaves have many glandular trichomes, which may be related to pest resistance. In order to collect the constituents of glandular trichomes, leaves were rinsed using dichloromethane (DCM) to obtain the rinsate, and the plant residues were subsequently extracted by DCM to obtain a DCM extract containing the internal constituents of yacón leaves. Biologic evaluations revealed that insect antifeedant activity was stronger for the rinsate than for the DCM extract against the common cutworm. The major constituents of rinsate were isolated by silica gel flash chromatography and were identified as sesquiterpene lactones (SLs), uvedalin (**1**) and enhydrin (**2**) and uvedalin aldehyde (**3**), collectively known as melampolides. Although SLs **1** and **2** exhibited remarkably strong insect antifeedant activity, SL **3** and reduced corresponding derivatives (**4** and **5**) of **1** and **2** exhibited moderate insect antifeedant activity. Additionally, the two analogs, parthenolide (**6**) and erioflorin (**7**) showed moderate insect antifeedant activity. The results indicate that the substituent patterns of SLs may be related to the insect antifeedant activities. The insect antifeedant activities of SLs **1** and **2** were similar to that of the positive control azadirachtin A (**8**), and thus these natural products may function in chemical defense against herbivores.

## 1. Introduction

The Asteroideae is a subfamily of the Asteraceae and includes the genera *Tanacetum*, *Senecio*, *Inula* and *Helianthus*. *Tanacetuem cinerariifolium* (Trevir.) Sch. Bip. syn. *Pyrethrum cinerariifolium* Trevir. produces sesquiterpene lactones (SLs) as well as pyrethroids [1,2]. It is well known that most of the Asteroideae produce SLs [3]. In general, SLs show a remarkable biologic activity toward various insect species [4,5,6,7].

Yacón (*Smallanthus sonchifolius* (Poepp. & Endl.) H. Rob) is a perennial crop plant of the Asteroideae, and its enlarged roots are used as a foodstuff (Figure 1). Since this plant exhibits high resistance to pests, it is easy to cultivate using small amounts of pesticides. Yacón has a large number of glandular trichomes on the abaxial surface of its leaves and this was presumed to function as one of its insect-resistance factors (Figure 2). However, yacón produces a large amounts of SLs and polyphenols, the aerial parts are sometimes used as functional foods [8,9,10].

Most trichomes are the aerial epidermal protuberances of plants, where they play a critical role as contributing to avoiding feeding damage by herbivores [11,12,13]. Trichomes are classified into hairy trichome or glandular trichome. Glandular trichomes consist of several cells and synthesizes biologic active compound. Moreover, then they store various chemicals, and various kinds of biologic active compounds in the glandular trichomes have been reported from various plant species [14]. In a function of glandular trichomes, a tomato cultivar without hairy trichomes produced large amounts of terpenoids compared with its cultivar with a hairy trichomes, and these terpenoids contribute to the herbivore resistance in tomato plants [15]. Additionally, plant trichomes can physically interfere with insect locomotion over the leaf surface and as a consequence avoid feeding damage from the herbivores [16]. In particular, most glandular trichomes synthesize and store various biologic active compounds with herbivore repellent and antimicrobial activities and they may provide a function as a chemical defense system for them. In our previous studies, the Asteraceae weed *Heterotheca subaxillaris* (Lam.) Britt. & Rusby was disclosed to produce and store sesquiterpene carboxylates in glandular trichomes on the leaf surface. These sesquiterpenes was found to exhibit phytotoxicity and insect antifeedant activity [16,17]. The Orobanchaceae weeds yellow bartsia (*Parentucellia viscosa* (L.) Caruel., syn. *Bartsia viscosa* L.) and Mediterranean linseed (*Bellardia trixago* (L.) All., syn. *Bartsia trixago* L.) also produce large amounts of exudates consisting of diterpenes for tiny insect entrapment on the trichomes of their leaf surface. The trichome density and insect antifeedant diterpene, kolavenic acid in glandular trichomes were both important for herbivore resistance of yellow bartsia [16].

The aim of present study was to unravel the resistant factors in yacón. In this study, we report the functional properties and SLs in glandular trichomes, from chemical ecological points of view.

## 2. Results and Discussion

The antifeedant activity evaluation showed that the rinsate had stronger insect antifeedant activity than dichloromethane (DCM) extract against common cutworms (*Spodoptera litura* F.) (Figure 3).

This plant species has many glandular trichomes on abaxial leaf surfaces (Figure 2). The rinsate contained the extract from their glandular trichomes, therefore the glandular trichomes may play a role in the defense system against phytophagous insect pests in this plant. The SLs uvedalin (**1**) and enhydrin (**2**) were found to exist in glandular trichomes on the abaxial leaf surfaces of yacón using matrix-assisted laser desorption/ionization mass spectrometry (MALDI-MS) imaging [18]. Similar results were reported for another member of the Asteroideae, Mexican sunflower (*Tithonia diversifolia* (Hermsl.) A. Gray), which also has glandular trichomes that produce SLs—the rinsate prepared from this plant showed insect antifeedant activity against a lepidopteran insect [19]. Camphorweed (*H. subaxillaris*), a weed from the Asteraceae, had a huge amount of glandular trichomes on the leaf surface in our previous studies. The rinsate prepared from this plant also showed strong insect antifeedant activity against common cutworms, and several calamenene-type sesquiterpene carboxylates were identified as insect antifeedants [16].

The major constituents of the rinsate prepared from aerial parts of yacón were identified as uvedalin (**1**) and its oxidative congener enhydrin (**2**) and uvedalin aldehyde (**3**), collectively known as melampolides [20] (Figure 4). It was presumed that these SLs prevent feeding damage by herbivores, but these natural products did not kill herbivores in the field (personal observation by M. Morimoto). Actually, the speed of common cutworms eating yacón leaves was very low in the field. Based on this field observation, these compounds were evaluated for insect antifeedant activity against common cutworms. The SLs **1** and **2** exhibited remarkably strong insect antifeedant activity and, excepting SL **3**, showed similar insect antifeedant activity to the positive control azadirachtin A (**8**). Thus, these melampolides may function in the yacón chemical defense system against herbivores. As seen in other structure–activity relationship (SAR) information on these melampolides, hydrogenation of the exomethylene on their g-lactone ring slightly decreased their biologic activity (Figure 4, Table 1). A similar case was reported concerning exomethylene on the g-lactone ring being critical for insect antifeedant activity of eudesmanolide from *Encelia* spp., also of the Asteraceae [21]. In general, since an exomethylene on the g-lactone ring will combine with an SH group of protein, this chemical moiety is the important to appear biologic activity. For the herbicidal guaianolide, its glutathione adduct (dehydrozaluzanin C) was also reported [22].

To determine the critical structural factor(s), the analog SLs **6** and **7** with different functional groups were evaluated for insect antifeedant activity. These SLs drastically decreased the biologic activity. These findings suggested that the g-lactone ring moiety was not essential for insect antifeedant activity in the tested SLs. The common structural features were hemiterpene, methylester and acetoxy groups, among the strong insect antifeedant SLs **1** and **2** and azadirachtin A (**8**) used as a positive control. Additionally, the exomethylene moiety on the g-lactone ring may have increased the biologic activity compared with hydrogenated derivatives (Table 1). As seen in other SAR information on these melampolides, the substituent patterns of SLs may be related to the appearance of the insect antifeedant activities (Figure 5, Table 1). Additionally, the two analogs, parthenolide (**6**) from *T. parthenum* L. and erioflorin (**7**) from *Helianthus strumosus* L., showed moderate insect antifeedant activity. Parthenolide (**6**) has no functional groups other than the g-lactone ring and epoxide and erioflorin (**7**) also has few substituents in addition to methacrylate ester and hydroxyl group. The SLs **6** and **7** showed moderate insect antifeedant activity against common cutworms. Thus, the substituent patterns of melampolides may be related to presence of these insect antifeedant activities. Interestingly, the g-lactone stereochemistry of SLs was also found to be related to herbivore resistance in the field for *Xanthium strumarium* L. (Asteraceae), which has glandular trichomes containing the SL xanthanolide [23] isomers of *cis*/*trans*-fused lactone differed in herbivore resistance potential, with the *trans*-fused form showing stronger resistance than the *cis*-fused form [24]. The insect antifeedant SLs **1–3**, from glandular trichomes of yacón leaves, were also in *trans*-fused form (Figure 4).

Consequently, SLs **1–3** from glandular trichomes of yacón leaves were presumed to play a role in chemical defense. The term “allelopathy” is defined an interaction between different species. Any chemical acting as an allelochemical should directly affect the species concerned, which in this study were yacón and common cutworms. In the case of plant–plant chemical interactions, when the constituents produced by the donor plant affect the receiver plant it is usually via soil and environmental media. In such cases, identifying and demonstrating an allelochemical is occasionally difficult. In our study, it was easy to determine that the SLs were allelochemicals because they provided yacón with herbivore resistance.

Finally, we evaluated the relationship between the insect antifeedant potential of SLs and herbivore resistance of tested plants using intact plant leaves. In this dual-choice type test for preference by tested phytophagous insects, the herbivore common cutworm consumed many more leaf disks prepared from fresh *H. strumosus* leaves compared with those prepared from fresh yacón leaves (Figure 6). The result seemed to be related to the insect antifeedant potential of constituents and herbivore resistance of tested plants—it was similar to the stronger insect antifeedant activity of SLs **1** and **2** extracted from yacón compared to SL **7** extracted from *H. strumosus* (Table 1).

## 3. Materials and Methods

### 3.1. General

The 1D- and 2D-NMR (COSY, HSQC and HMBC) spectra were obtained using an Avance 400 instrument (Bruker Co. Ltd., Bremen, Germany) with solvent signal as an internal reference. The EI-MS spectra were measured using a JMS-K9 (JEOL Co. Ltd., Tokyo, Japan). A low-vacuum scanning electron microscope (LVSEM) was used for observation of leaf surfaces by SU-1510 (Hitachi High-Technologies, Tokyo, Japan). Optical rotations were measured with a high-sensitivity SEPA 300 polarimeter (Horiba). Flash column chromatography was performed on an Isolera One (Biotage, Uppsala, Sweden).

### 3.2. Chemicals and Insects

All naturally occurring SLs (**1–3**) were isolated from DCM rinsate prepared from the aerial parts of fresh yacón leaves cultivated and collected in Ikoma City, Nara Pref., Japan. The corresponding derivatives (**4**, **5**) were prepared by organic synthesis [25]. Erioflorin (**7**) was isolated from DCM rinsate prepared from the aerial parts of *H. strumosus* cultivated and collected in Ikoma City.

The authentic reference compounds, parthenolide (>97%) (**6**) and azadirachtin A (Aza) (>95%) (**8**), were purchased from Tokyo Kasei Kogyo Co. Ltd., Japan and Sigma-Aldrich Co., LLC, UK, respectively.

Common cutworms (*S. litura* F., Lepidoptera: Noctuidae) were purchased from Sumika Technoservice (Takarazuka City, Japan). The insects were reared on an artificial diet (Insect LF, Nihon Nosan Kogyo Co. Ltd., Japan) in a controlled environment at 26.5 °C and 60% humidity.

### 3.3. Plant Materials

Both yacón (*S. sonchifolius*) and *H. strumosus* were cultivated and harvested in Ikoma City during 2017–2019. The *H. tuberosus* was collected from the riverside of Tomio River in Nara City in 2019. These mature plants were cultivated at Nov. in these years.

### 3.4. LVSEM Observation

The plant specimens were prepared as a small piece from fresh yacón leaf. The specimens were mounted using double-sided carbon tape. For LVSEM, SU-1510 (Hitachi High-Technologies, Tokyo, Japan) was used with the vacuum condition and accelerating voltage set at 30 Pa and 15 kV, respectively. The effect of DCM rinsed within a few seconds was measured both for intact (a) and treated (b) leaves (Figure 7).

### 3.5. Extraction and Isolation

DCM rinsates were prepared from the fresh aerial parts of yacón (4.7 kg) by rinsing using DCM solvent (4 L) for ca. 10 s. The DCM solution obtained was concentrated under reduced pressure to obtain a DCM rinsate (9.5 g, 0.20% yield). Then the plant residues of the aerial parts of yacón were extracted using DCM for three days. After filtration, the extract was concentrated under reduced pressure to obtained DCM extract (12.2 g, 0.26% yield). The constituents (**1–3**) were isolated from DCM rinsates by silica gel flush column chromatography with hexane and ethyl acetate as eluent [20], and the corresponding hydrogenated derivatives (**4**, **5**) were prepared using hydrogenation with Pd–C as catalyst [26]. DCM rinsates were prepared from the fresh aerial parts of *H. strumosus* (0.75 kg) by rinsed the plant using DCM solvent (2 L) for ca. 10 s. The DCM solution obtained was concentrated under reduced pressure to obtain a DCM rinsate (1.0 g, 0.14% yield). The constituent (**7**) was isolated from DCM rinsates by silica gel flush column chromatography with hexane and ethyl acetate as eluent [27].

### 3.6. Insect Antifeedant Biologic Assay (Dual-choice Type Test)

Test compounds and extracts of the insect antifeedant activity were evaluated against common cutworm (*S. litura* F.). Using leaf disks of 2-cm diameter were prepared using a cork borer from fresh sweet potato (*Ipomoea batata* cv. Narutokintoki) leaves that was cultivated at the farm in Nara campus of Kindai University (Nara, Japan). The upper surface of two leaf disks were treated with test compound in acetone and two other leaf disks were treated with acetone only as a control. After the acetone had completely evaporated at room temperature. These four leaf disks were placed in alternating positions in the same 9.5-cm-diameter glass Petri dish with moistened filter paper on the bottom. The reference dish consisted of only four control leaf disks. Then, 10 larvae (third instar) were released into the dish and allowed to feed. The dish was kept in an insect rearing room at 26.5 °C in darkness for 5–6 h. Each experiment was terminated when the consumption of four control leaf disks in the reference dish was completed. The consumed leaf disks were digitized using a PC scanner. Data were analyzed on a PC using NIH Image J, U. S. National Institutes of Health, Bethesda, Maryland, USA (http://rsb.info.nih.gov/ij/index.html). For each experiment, the data for an intact disk were measured and compared with those of a treated disk. To measure the activities of test compounds, we used an antifeedant index: AFI = (% of treated disks consumed / (% of treated disks consumed + % of control disks consumed)) × 100. The AFI value was converted to the inhibitory rate (%): inhibitory rate (%) = (50 − AFI) × 2. The insect antifeedant potency of test compounds was evaluated in terms of the median effective dose (ED_50_) value for the rate of feeding inhibition calculated from the area of the leaf disk consumed [28]. A straight line was fitted to the points obtained using the bioassay, and the ED_50_ calculated as the dose corresponding to the midpoint between complete inhibition and no effect using the probit method [29] using R software (ver. R. 2.12.1, R Foundation for Statistical Computing); the 95% confidence interval (95% CI) was also determined.

## 4. Conclusions

The isolated SLs from glandular trichomes of yacón showed strong insect antifeedant activity against common cutworms, thus may function as a kind of the resistant factors in this plant. The SLs, uvedalin (**1**) and its oxidative congener, enhydrin (**2**), known as melampolide were major constituent and showed remarkable insect antifeedant activity as well as the positive control azadirachtin A (**8**). Additionally, the two analogs, parthenolide (**6**) and erioflorin (**7**) showed moderate insect antifeedant activity against common cutworms. As a result, the substituent patterns of SLs maybe concerned with appearance of insect antifeedant activity. These results suggested that the existence of SLs in glandular trichomes was important to show phytophagous insect resistance of yacón.

## Figures and Tables

**Figure 1 plants-09-00848-f001:**
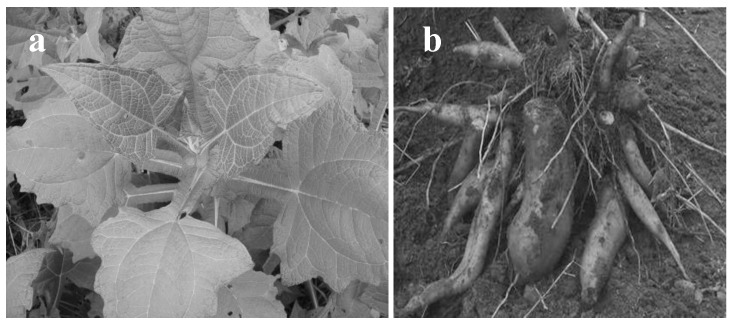
Yacón (*S. sonchifolius* (Poepp. & Endl.), H. Rob): (**a**) aerial parts and (**b**) root parts.

**Figure 2 plants-09-00848-f002:**
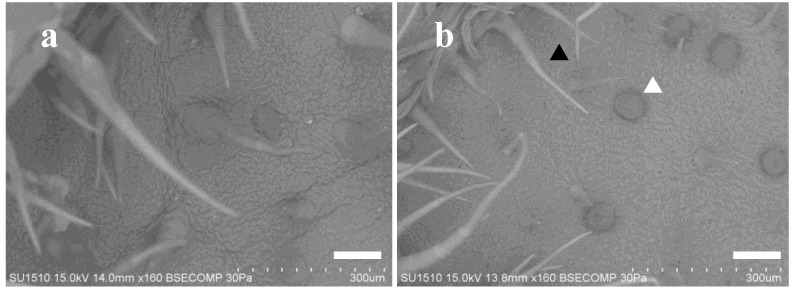
Low-vacuum scanning electron microscope (LVSEM) images of hairy (black triangle) and glandular (white triangle) trichomes on the yacón leaf surface (×160). (**a**) adaxial; (**b**) abaxial (bar: 100 µm).

**Figure 3 plants-09-00848-f003:**
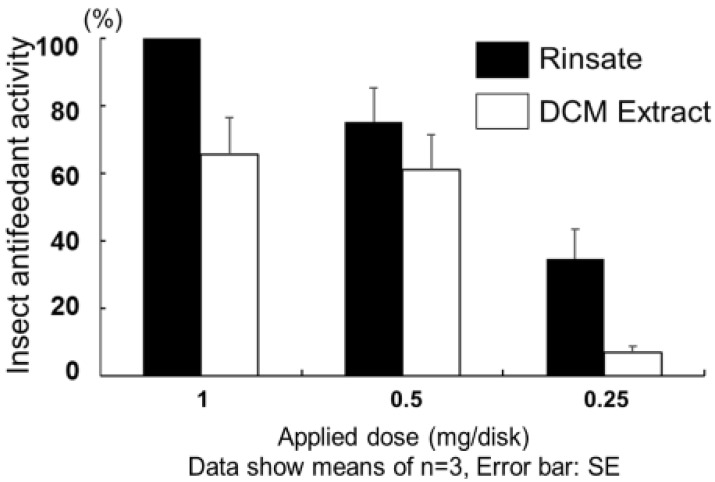
Insect antifeedant activity of rinsate and dichloromethane (DCM) extract from aerial parts of yacón.

**Figure 4 plants-09-00848-f004:**
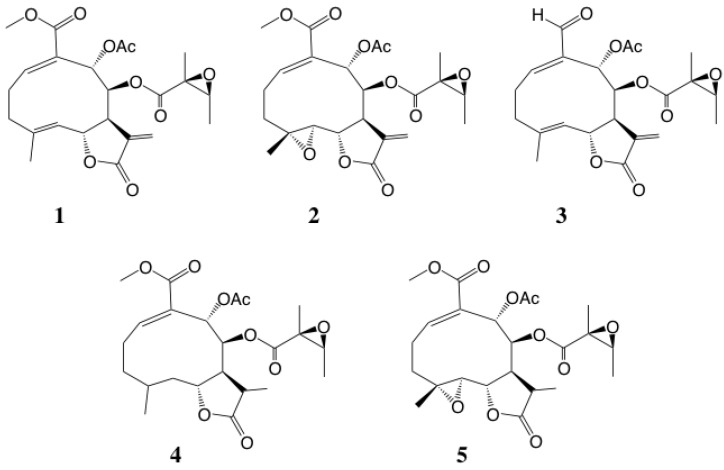
Melampolides and derivatives **1–5** from yacón leaf trichomes.

**Figure 5 plants-09-00848-f005:**
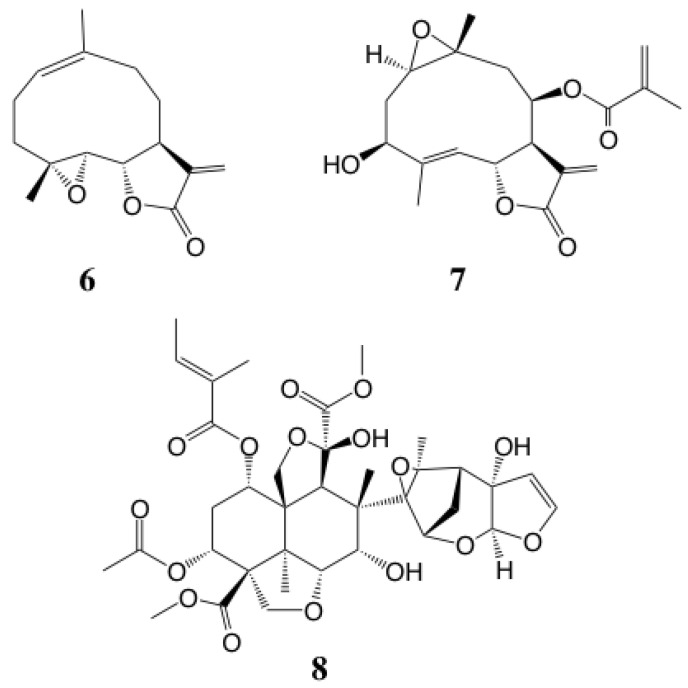
Melampolide analogs **6**, **7** and azadirachtin A (**8**) used for the evaluation of insect antifeedant activity.

**Figure 6 plants-09-00848-f006:**
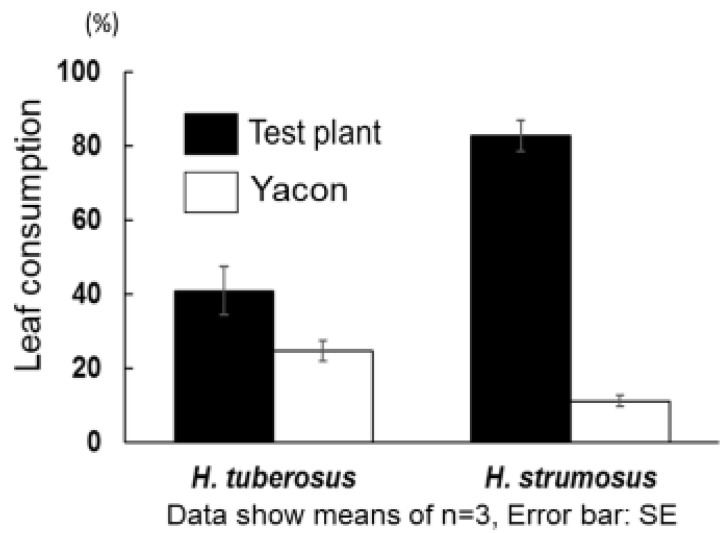
Comparison of herbivore resistance between yacón leaf and *H. tuberosus* or *H. strumosus* leaf.

**Figure 7 plants-09-00848-f007:**
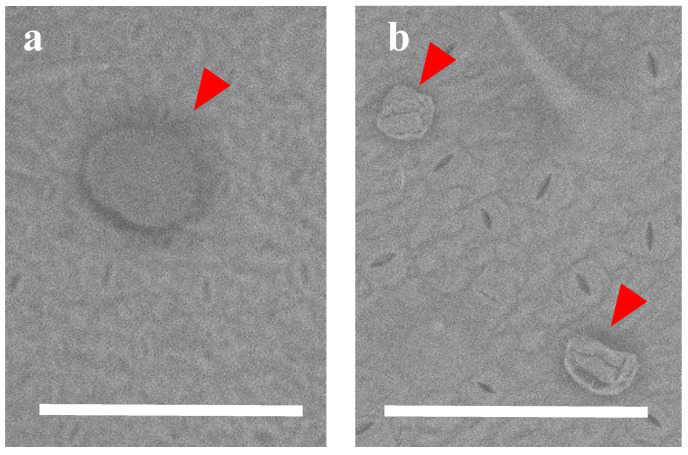
LVSEM images of glandular trichomes on the yacón leaf surface (**a**) before DCM rinse and (**b**) within a few seconds after DCM rinse (bar: 100 µm).

**Table 1 plants-09-00848-t001:** Insect antifeedant activity of tested sesquiterpene lactones and azadirachtin A against common cutworm (*S. litura*) third-instar larvae.

	Antifeedant Activity against *S. litura* Larvae		
Compounds	ED_50_ (mg/cm^2^)	ED_50_ (µmol/cm^2^)	95% CI (mg/cm^2^)
**1**	0.0080	0.018	0.0062–0.012
**2**	0.0042	0.0090	0.0026–0.0086
**3**	0.030	0.072	0.022–0.043
**4**	0.040	0.058	0.015–0.070
**5**	0.027	0.088	0.0076–0.053
**6**	0.25	7.3	0.15–0.69
**7**	0.20	7.9	0.18–0.21
**8 (Aza)**	0.00023	0.00032	0.00021–0.00027

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
