# Peer review of "Chemical Defense of Yacón (Smallanthus sonchifolius) Leaves against Phytophagous Insects: Insect Antifeedants from Yacón Leaf Trichomes"

_plants, 2020, doi:10.3390/plants9070848_

Round 1

Reviewer 1 Report

The presented manuscript contains interesting data on the antifeedant activity of natural sesquiterpene lactones of Smallanthus sonchifolius against Spodoptera litura. However, there are certain issues that have to be cleared before the manuscript is considered for publication.

  1. The first issue is terminology:

a) ‘This plant shows highly tolerance toward its pests. As a resistance factor...’ Please, stick to the generally accepted definitions of resistance mechanisms. Tolerance is one of resistance mechanisms but it is not based on allelochemicals. Tolerance is the ability of the plant to overcome the effects of herbivory. In the case of allelochemicals, we can consider antixenosis and/or antibiosis. Also the use of term ‘allelopathy’ (line 140) does not fit here.

b) Asteraceae, Asteroideae – (lines 87-90) – these are different taxonomical units, please chose one, or use the expression: Asteraceae: Asteroideae. Asteroideae is the subfamily of Asteraceae.

c) what is the ‘feeding speed’ of common cutworms (line 98)?

  1. The second issue is the proper explanation of the aims of the study. There is no statement of the aim of the study. It should appear as the last paragraph of the Introduction. It seems that the goal was to establish the role of the studied chemicals as antifeedants and the role of the structure of the molecules in the expression of the activity. It must be stated clearly. Otherwise, it is not possible to judge whether the goals of the study were achieved or not. Why the experimental insect was Spodoptera litura? Is it the pest of Smallanthus sonchifolius? Please, explain it in the Introduction. The background of the study is too brief. Please, say more about the plant-insect system you are studying.
  2. The third issue is the English language. It certainly must be improved. Although the text is understandable, there are plenty of grammatical errors.
  3. Other remarks

Introduction: First paragraph – the abbreviations should be explained in full.

Results and Discussion: There are no indications of significant differences in the figures and in the tables. What was the statistical analysis? There is no mention about it in the text. It is crucial for the conclusions about both the antifeedant activity and the structure-activity aspect.

How was the insect antifeedant activity of tested sesquiterpenolides and azadirachtin A evaluated? In the dual choice test? How the activities were compared? What statistical test?

Methods: The use of acetone – acetone is toxic to plant tissues. Did the Authors consider this?

Number of replicates: 3 (three) is really too low to evaluate the activity of the antifeedant.

Author Response

Response to reviewers,

We appreciate that you gave several valuable suggestions to our study. All of reviewers pointed the English matter. I rewrote through all part of my paper.

And I improved and changed pointed parts in our paper.

----------------------------------------------------------------------------

1:Comments and Suggestions for Authors

The presented manuscript contains interesting data on the antifeedant activity of natural sesquiterpene lactones of Smallanthus sonchifolius against Spodoptera litura. However, there are certain issues that have to be cleared before the manuscript is considered for publication.

  1. The first issue is terminology
  2. a) ‘This plant shows highly tolerance toward its pests. As a resistance factor...’ Please, stick to the generally accepted definitions of resistance mechanisms. Tolerance is one of resistance mechanisms but it is not based on allelochemicals. Tolerance is the ability of the plant to overcome the effects of herbivory. In the case of allelochemicals, we can consider antixenosis and/or antibiosis. Also the use of term ‘allelopathy’ (line 140) does not fit here.

I am considering that the resistant factor of a plant produced the biological active compound, ex. insecticidal natural product, is allowed to refer as an allelopathy against herbivores. The definition of allelopathy is chemical effects between different species, plant and plant, plant and insect, plant and microorganism and so on. The chemicals related with these phenomena are referred allelochemical. In my paper, though the SLs act as defense chemical of yacon plant, these SLs are allelochemicals. Also tolerance is including to phenomena of allelopathy in a broad sense I think.

  1. b) Asteraceae, Asteroideae ? (lines 87-90) ? these are different taxonomical units, please chose one, or use the expression: Asteraceae: Asteroideae. Asteroideae is the subfamily of Asteraceae.

I added the explanation of subfamily for this part.

  1. c) what is the ‘feeding speed’ of common cutworms (line 98)?

I meant the speed of consumption by common cutworms. I changed the words "the speed of common cutworms eating yacon leaves".

  1. The second issue is the proper explanation of the aims of the study. There is no statement of the aim of the study. It should appear as the last paragraph of the Introduction. It seems that the goal was to establish the role of the studied chemicals as antifeedants and the role of the structure of the molecules in the expression of the activity. It must be stated clearly.

I added this sentence to end of introduction "This paper showed that SLs from glandular trichomes of yacon leaves maybe played a critical role in chemical defense.

Otherwise, it is not possible to judge whether the goals of the study were achieved or not. Why the experimental insect was Spodoptera litura? Is it the pest of Smallanthus sonchifolius? Please, explain it in the Introduction. The background of the study is too brief. Please, say more about the plant-insect system you are studying.

I observed that S. litura fed the yacon leaves in yacon cultivated field. (Line 99)

And this worms is also very common to use for insect antifeedant evaluation.

  1. The third issue is the English language. It certainly must be improved. Although the text is understandable, there are plenty of grammatical errors.

I improved.

  1. Other remarks

Introduction: First paragraph ? the abbreviations should be explained in full.

I improved.

Results and Discussion: There are no indications of significant differences in the figures and in the tables. What was the statistical analysis? There is no mention about it in the text. It is crucial for the conclusions about both the antifeedant activity and the structure-activity aspect.

In this study, the intensity of tested compounds using ED50 value calculated by Probit analysis. I think that this is enough to compare with test compounds.

How was the insect antifeedant activity of tested sesquiterpenolides and azadirachtin A evaluated? In the dual choice test? How the activities were compared? What statistical test?

The comparison of insect antifeedant activity in test compounds based on the ED50 value calculated from liner regression analysis.

Methods: The use of acetone ? acetone is toxic to plant tissues. Did the Authors consider this?

Actually, acetone is not safe against plant tissue but this effect was very low. Because the carrier solvent used acetone was readily evaporated when the actone solution was applied to leaf disk.

Number of replicates: 3 (three) is really too low to evaluate the activity of the antifeedant.

I agree that the large number of replicate is better but unstable point was done by the several trials in this test. Now I am keeping that the three replicates is an our standard to calculated to the average value. Other researcher adapted three replicates, Azucena González-Coloma group in Plants 2019, 8 176.

Reviewer 2 Report

The first time an abbreviation is used the full name must be given. It is not enough to write them in the introduction.

Lines 34-38 - The first paragraph is barely understandable. Please rewrite.

 Line 37 - sesquiterpenolids (SLs) - please always give the full name and abreviations in brackets the first time you use them

I must confess that reading this paper was not easy. The English is quite rough and the description of results and procedures are quite difficult to follow. For instance in the MM section the order should be changed according to the procedures: first plant material followed by extraction, then isolation and identification and insects interactions. At present this section is quite difficult to follow.

The authors extract the content of the trichomes and then they use the same solvent for the three days. It is obvious that most of the chemicals implied in plant defence were already taken away.

Results and discussion are not clear. There is a lot of mix between theoretical discussion and actual results.

I would suggest the authors to rewrite the paper.

Author Response

Response to reviewers,

We appreciate that you gave several valuable suggestions to our study. All of reviewers pointed the English matter. I rewrote through all part of my paper.

And I improved and changed pointed parts in our paper.

----------------------------------------------------------------------------

2:Comments and Suggestions for Authors

The first time an abbreviation is used the full name must be given. It is not enough to write them in the introduction.

I added the full spelling for the abbreviation.

Lines 34-38 - The first paragraph is barely understandable. Please rewrite.

I added the explanation of subfamily for this part.

 Line 37 - sesquiterpenolids (SLs) - please always give the full name and abreviations in brackets the first time you use them

I added the full spelling for the abbreviation.

I must confess that reading this paper was not easy. The English is quite rough and the description of results and procedures are quite difficult to follow. For instance in the MM section the order should be changed according to the procedures: first plant material followed by extraction, then isolation and identification and insects interactions. At present this section is quite difficult to follow.

I understood you commented, but this order in MM section is adequately for us. I want to keep this order this time.

The authors extract the content of the trichomes and then they use the same solvent for the three days. It is obvious that most of the chemicals implied in plant defence were already taken away.

We evaluated this matter but unknown insect antifeedants maybe exist in DCM extract, I guess. Now we are continuing to looking for(figure 3).

Results and discussion are not clear. There is a lot of mix between theoretical discussion and actual results.

I would suggest the authors to rewrite the paper.

I have improved all part of this paper to easy to understand by readers.

----------------------------------------------------------------------------

Reviewer 3 Report

Introduction needs to start with a more statement of the problem. Define SL at first use in the text.

l.41: tolerance or resistance? these concepts needs to be distinguished.

l.43: 'to function as resistence factors'

l.55-57: unclear. This sentence seems to indicate that the trichome-free mutant was more herbivore-resistant than the mutant with trichomes, presumably the opposite of what was intended.

Fig. 2: what is meant by 'hairy and glandular trichomes'? are there two sorts of trichomes in the image? I only see simple trichomes. No evidence that these trichomes are glandular.

More information is needed about the plants from which SLs were collected: mature or expanding leaves? how were rinsates prepared? were leaves flash-frozen or processed fresh?

Fig. 7: is this the adaxial or abaxial leaf surface? scale is required on the image.

Author Response

Response to reviewers,

We appreciate that you gave several valuable suggestions to our study. All of reviewers pointed the English matter. I rewrote through all part of my paper.

And I improved and changed pointed parts in our paper.

3:Introduction needs to start with a more statement of the problem.

We are trying to disclose a part of the phenomena in nature. I think that I do not need the problem statement in this paper. ex. environmental pollutions and development of ecological friendly pesticides: botanical pesticides.

Define SL at first use in the text.

I added the full spelling for the abbreviation.

l.41: tolerance or resistance? these concepts needs to be distinguished.

Tolerance is endure from feeding damage, Resistance is active toward pests using by weapons.

l.43: 'to function as resistence factors'

I changed as " to function as one of its insect-resistance factors"

l.55-57: unclear. This sentence seems to indicate that the trichome-free mutant was more herbivore-resistant than the mutant with trichomes, presumably the opposite of what was intended.

This part was explained that the trade off of tomato plants. No trichome tomato have to be made toxic defense chemical for the avoidance from herbivore damage, instead of production of trichome under the same cost.

Fig. 2: what is meant by 'hairy and glandular trichomes'? are there two sorts of trichomes in the image? I only see simple trichomes.

I added two triangles on the fig 2 (black and white, hairy and glandular respectively)

No evidence that these trichomes are glandular.

The glandulars were washed out by DCM rinse (fig.7). Because almost glandular trichome consist of lipophilic chemicals.

More information is needed about the plants from which SLs were collected: mature or expanding leaves?

I added the sentence "These mature plants were cultivated at Nov. in these years.

" at end of section 3.3.

how were rinsates prepared? were leaves flash-frozen or processed fresh?

I noted section 3.5.

Fig. 7: is this the adaxial or abaxial leaf surface? scale is required on the image.

I added the scales on Fig.2 and 7.

Round 2

Reviewer 1 Report

The manuscript was improved but there remain certain issues to be cleared.

The terminology used: although I can accept the use of ‘allelopathy’ in describing plant-insect interactions (indeed sometimes the term is used in this very broad sense), the use of ‘tolerance’ is not correct. I refer the Authors to the publication by Smith C. 2005.Plant resistance to arthropods: molecular and conventional approaches. NewYork: Springer;  Koch et al. 2016 https://doi.org/10.3389/fpls.2016.01363; Mitchell et al. 2016 (https://doi.org/10.3389/fpls.2016.01132); Peterson et al. (2017), PeerJ, DOI 10.7717/peerj.3934 16. In these works you will find the definitions of antibiosis, antixenosis and tolerance, which are the three forms of plant resistance to herbivores. Your research refers most likely to antixenosis. Not to tolerance. You did not study plant metabolism in response to herbivory.

Other remarks:

line 41 – reference missing

line 58 – repellent

lines 66-68 – this is the conclusion of the study and not the aim. Please, state the aim of the study clearly: ‘The aim of the present study was.....’ The statement should correspond with the experiments presented in the manuscript. Apart from this statement, one sentence is needed to show how the aim was to be reached, that is what particularly the authors investigated. This should be a separate paragraph of the introduction.

lines 83-84: The sentence ‘Although the rinsate contained the extract from their glandular trichomes, the glandular trichomes may play a role in the defense system against phytophagous insect pests in this plant.’ should probably mean: ‘The rinsate contained the extract from their glandular trichomes, therefore the glandular trichomes seem to play a role in the defense system against phytophagous insect pests in this plant’. The original version states otherwise but when one sees the figure 3, the meaning is in the contrary.

line 99: perhaps you should add ‘(personal observation by xx)’ at the end of the sentence. You did not put any results on the consumption rate inthe manuscript.

lines 141-144 – why define allelopathy in this place? It is out of the context here.

lines 174-177 – please add the information on the synthesis – if it was described in any publication, please add the reference. If it has never been described, please do it now.

Please, add the ‘Conclusions’ section at the end of the manuscript to point exactly what resulted from the study. Now, the information is dispersed in the Results and Discussion section, therefore difficult to judge what the Authors did and what was done by the others or in the Authros’ previous studies. This should correspond with the Aims of the study that should be stated at the end of the introduction.

Author Response

1.

Comments and Suggestions for Authors

The manuscript was improved but there remain certain issues to be cleared.

The terminology used: although I can accept the use of ‘allelopathy’ in describing plant-insect interactions (indeed sometimes the term is used in this very broad sense), the use of ‘tolerance’ is not correct. I refer the Authors to the publication by Smith C. 2005.Plant resistance to arthropods: molecular and conventional approaches. NewYork: Springer;  Koch et al. 2016 https://doi.org/10.3389/fpls.2016.01363; Mitchell et al. 2016 (https://doi.org/10.3389/fpls.2016.01132); Peterson et al. (2017), PeerJ, DOI 10.7717/peerj.3934 16. In these works you will find the definitions of antibiosis, antixenosis and tolerance, which are the three forms of plant resistance to herbivores. Your research refers most likely to antixenosis. Not to tolerance. You did not study plant metabolism in response to herbivory.

I greatly appreciate your kindly good advise for my understanding of terminology about them. I read several papers including your recommending three papers, and I understood you mentioned.

So I change the term tolerance to resistance in our paper.

Other remarks:

line 41 – reference missing

I changed as have been suggested to --------> was presumed to

line 58 – repellent

I changed.

lines 66-68 – this is the conclusion of the study and not the aim. Please, state the aim of the study clearly: ‘The aim of the present study was.....’ The statement should correspond with the experiments presented in the manuscript. Apart from this statement, one sentence is needed to show how the aim was to be reached, that is what particularly the authors investigated. This should be a separate paragraph of the introduction.

I added sentence and it was separated from introduction.

lines 83-84: The sentence ‘Although the rinsate contained the extract from their glandular trichomes, the glandular trichomes may play a role in the defense system against phytophagous insect pests in this plant.’ should probably mean: ‘The rinsate contained the extract from their glandular trichomes, therefore the glandular trichomes seem to play a role in the defense system against phytophagous insect pests in this plant’. The original version states otherwise but when one sees the figure 3, the meaning is in the contrary.

You are correct. I changed. Thank you.

line 99: perhaps you should add ‘(personal observation by xx)’ at the end of the sentence. You did not put any results on the consumption rate in the manuscript.

I agree. I inserted (personal observation by M. Morimoto)

lines 141-144 – why define allelopathy in this place? It is out of the context here.

This definition of allelopathy is followed to the term "allelochemical". It is better to explain allelopathy before appear the term, allelochemical, I think.

lines 174-177 – please add the information on the synthesis – if it was described in any publication, please add the reference. If it has never been described, please do it now.

I inserted the citation.

Please, add the ‘Conclusions’ section at the end of the manuscript to point exactly what resulted from the study. Now, the information is dispersed in the Results and Discussion section, therefore difficult to judge what the Authors did and what was done by the others or in the Authros’ previous studies. This should correspond with the Aims of the study that should be stated at the end of the introduction.

I made conclusion part.

Reviewer 2 Report

The authors did some improvement to the text. They refused several requests from the reviewers. It may still be improved.

Author Response

Comments and Suggestions for Authors

The authors did some improvement to the text. They refused several requests from the reviewers. It may still be improved.

I appreciate for you spend the time for improvement of our paper.

I tried to improve based on reviewers comment last time. And I also used English checking agent to improve English grammar, last time. And this time I did same manner. I hope you will see this matter.

Round 3

Reviewer 1 Report

The manuscript has been improved and the missing information was added.

I am satisfied with the changes.

Reviewer 2 Report

This is the third review of the paper. I do not think that the  authors fullfilled all the tasks required by the reviewers.